# Beyond Winning and Losing: Modeling Human Motivations and Behaviors with Vector-valued Inverse Reinforcement Learning

## Abstract

In recent years, reinforcement learning methods have been applied to model game-play with great success, achieving super-human performance in various environments, such as Atari, Go and Poker. However, those studies mostly focus on winning the game and have largely ignored the rich and complex human motivations, which are essential for understanding the agents' diverse behavior. In this paper, we present a multi-motivation behavior modeling which investigates the multifaceted human motivations and models the underlying value structure of the agents. Our approach extends inverse RL to the vectored-valued setting which imposes a much weaker assumption than previous studies. The vectorized rewards incorporate Pareto optimality, which is a powerful tool to explain a wide range of behavior by its optimality. For practical assessment, our algorithm is tested on the World of Warcraft Avatar History dataset spanning three years of the gameplay. Our experiments demonstrate the improvement over the scalarization-based methods on real-world problem settings.

## 1 Introduction

Reinforcement learning methods have been intensively applied to game environments with great success. Several landmark research works have been conducted on the games such as such as Atari, Go, and Poker where RL algorithms achieve super-human performance (Mnih et al. (2015); Silver et al. (2017); Heinrich & Silver (2016)). However, the majority of the studies focus on winning the game or achieving high scores, thus have largely ignored the rich and complex human motivations. The understanding of the motivation and the corresponding reward mechanism, which is essential for computing intelligence, has hence long been open. In fact, extending the single, scalarized reward function to model multifaceted motivations is non-trivial. The vector-valued reward does not describe the task as clear and succinct as the scalar reward does, and consequently, the optimality of the policy can be ill-defined. We argue that the extension instead imposes weaker assumption on the optimality of the policies, hence enable the model to analyze the diverse behavior from the perspective of the reward mechanism. It is shown to be equivalent to allow arbitrary scalarization function to process the multi-dimensional reward signals with the minimal assumption that the function is monotonic, as desired in analyzing the agents' behavior in the real world.

Previous studies have been discussing the setting where multiple reward signals are available. For example, features are commonly used in IRL since its inception (Ng et al. (2000); Abbeel & Ng (2004)), where features are treated as oracles and the reward is a function of those features. It was assumed that the combination of features is linear and was later extended to the nonlinear cases such as neural networks e.g. Wulfmeier et al. (2015); Finn et al. (2016)) and Gaussian processes e.g. Levine et al. (2011). In general, the scalarization-based methods do not incur violation of model assumptions if there exists a scalarized reward function (whether observed or not). However, as the environment is getting more complex and the motivations of the agents become diverse, the uniqueness of the scalar reward does not hold. In this context, we design the algorithm with weaker assumptions as desired.

In this paper, we propose a novel method called multi-motivation behavior modeling (MMBM) which takes the multifaceted, complex, and diverse motivations into consideration. Instead of finding

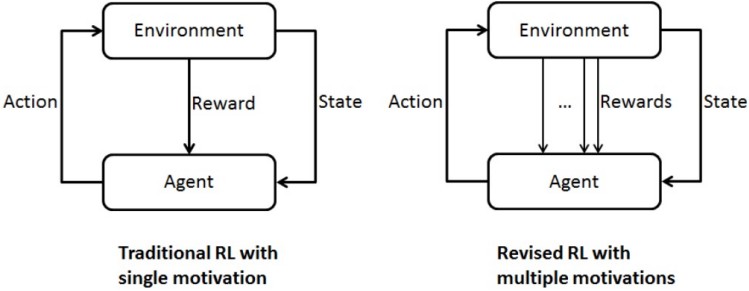

Figure 1: In the typical RL model (left), an agent has only one single motivation and maximizes one scalar reward. In MMBM (right), an agent or player has multiple motives and the goal is to optimize the combination of different rewards based on the agent's own value structure.

Table 1: Components of game motivation

| Components | Sub-components |
| --- | --- |
| Achievement | Advancement, Mechanics, Competition |
| Social | Socializing, Relationship, Teamwork |
| Immersion | Discovery, Role Playing, Customization, Escapism |

the scalarization in the reward space, we regard any policy that is not dominated by an alternative feasible policy as an optimal policy, as shown in Fig. 1. We build our analysis upon the theorem that the policy is not dominated if and only if it is optimal under a linear scalarization to connect the set of optimal policy with the simplex of the coefficient vector. In this way, we characterize the range of the value function under optimal policies as a convex hull. We further estimate the range via value learning on the extreme points and define the suboptimality of a policy as the distance between its corresponding value function and the range. As a result, we design an efficient algorithm with a linear program to minimize the distance and find the reward mechanism.

A significant advantage of MMBM is that it utilizes only off-policy learning: Both the range of the value function and the linear program depend on only the trajectories and does not require an access to the state dynamic transition. In this way, the algorithm is applicable to historical datasets such as data from online games. MMBM is tested on the World of Warcraft Avatar History (WoWAH) dataset Lee et al. (2011), which records the movement of players over a three-year period. We use the motivation theory of gameplay (Yee (2006), see Table 1) as the oracle of the vector-valued reward, and compare the analysis of the motivations with previous knowledge-based studies on WoW (Ducheneaut et al. (2006); Nardi & Harris (2006)). We further show the significant improvement on the inverse learning error over existing studies.

## 2 PRELIMINARIES

### 2.1 VECTOR-VALUED MARKOV DECISION PROCESS

We adopt the vector-valued Markov decision process (MDP) introduced by Wakuta (1995). We follow the convention that for two vectors $r, r' \in \mathbb{R}^d$, $r > r'$ if and only if $e_i^T r > e_i^T r$ for all $i \in [d]$, where $e_i^T$ has its $i$-th element equal to 1 and 0 otherwise. We define $r < r'$, $r \geq r'$, and $r \nleq r'$ similarly. The vector-valued infinite-horizon MDP is characterized by the finite state space $\mathbb{S}$, the finite action space $\mathbb{A}$, the transit probability $\mathbb{P}(s'|s,a)$, the reward signal $r = r(s,a) \in \mathbb{R}^d$, and the discount factor $\gamma$. At each step of the process, the agent chooses a feasible action $a \in \mathbb{A}$ and the system's state $s$ transits to $s'$ with probability $\mathbb{P}(s'|s,a)$, while the agent receives an immediate reward of $r(s,a,s')$. To describe the problem, we first define the value function $v^\pi(s) \in \mathbb{R}^d$ with respect to a fixed policy

$\pi$ by

$$v^{\pi}(s) = \mathbb{E}_{s' \sim \mathbb{P}, s \sim \rho_0(s)} \left[ \sum_{t=0}^{\infty} \gamma^t r_t | s_0 = s, \pi \right], \tag{1}$$

where $\pi$ chooses actions at each state, either deterministically or stochastically, also without ambiguity we write $r_t = \mathbb{E}_{s'_t}[r(s_t, a_t, s'_t)]$ and let $s_0$ be the initial state. We will omit the distribution terms $s' \sim \mathbb{P}, s_0 \sim \rho_0(s)$ in the rest part of the discussion. The objective of the agent is to find a policy $\pi$ such that

$$v^{\pi}(s_0) \not< v^{\pi'}(s_0) \tag{2}$$

holds for every feasible policy $\pi'$. Note that the objective of scalar-valued MDP is the special case where $r(\cdot) \in \mathbb{R}$ and $d = 1$. To characterize the optimal policies, we first denote $\Pi$ the set of policies that satisfy the condition equation 2. Also define $V(s) = \{v^{\pi}(s), \pi \in \Pi\}$ to be the set of value vectors that are only smaller or equal than ($\leq$) themselves in the range of the value function. $V(s)$ is also denoted as the Pareto frontier of the range of the value function Van Moffaert & Nowé (2014).

We describe the properties of $\Pi$'s elements. Similar to the recent studies in multi-object MDPs (Vamplew et al. (2008); Van Moffaert & Nowé (2014); Jaderberg et al. (2016); Mossalam et al. (2016)), we define the scalarization of the reward signal as

$$\tilde{r} = \phi^T r(s, a) \tag{3}$$

where $\phi \in \Phi \subset \mathbb{R}^d$ is the weight vector of the scalarization and $\Phi = \{\phi \in \mathbb{R}^d, \|\phi\|_1 = 1, \phi \geq 0\}$ is the simplex. We then define the set $\Pi_{\phi}$ of the optimal policies under the scalarized reward signal $\Pi_{\phi} = \arg\max_{\pi} \mathbb{E}[\sum_{t=0}^{\infty} \gamma^t \phi^T r_t | s_0, \pi]$. The following theorems shows the connection between the optimality of policies under the vectored reward and the optimality of policies under the scalarized reward.

**Theorem 1.** *For any policy $\pi$, $\pi \in \Pi$ if and only if there exists a vector $\phi \in \Phi$ such that $\pi \in \Pi_{\phi}$.*

**Theorem 2.** *There exists finite number of policies $\pi_1, ..., \pi_m$, such that the range of the value function $v^{\pi}(s)$ with respect to $\pi$ is the convex hull with $v^{\pi_1}(s), ..., v^{\pi_m}(s)$ being all its extreme points.*

Theorem 1 maps the optimal policies to the reward weight space $\Phi$ which is a standard simplex. Theorem 2 then maps the corresponding values to a convex hull based on which the suboptimality measure can be established. Both theorems are important to us and our analysis is build upon them. We refer the readers to Wakuta (1995) for both the proofs.

## 2.2 INVERSE REINFORCEMENT LEARNING

Inverse reinforcement learning (IRL) reverses the input and output pairs of RL algorithms, computing the rewards function according to the policies or trajectories of the agents. The topic has been intensively studied and developed since its inception (Ng et al. (2000)), including max-margin based methods (Abbeel & Ng (2004); Ratliff et al. (2006); Syed & Schapire (2008); Neu & Szepesvári (2012)), max-entropy IRL (Ziebart et al. (2008); Boularias et al. (2011); Finn et al. (2016)), Bayesian IRL (Ramachandran & Amir (2007); Lopes et al. (2009); Levine et al. (2011); Michini & How (2012)) and etc. The initial investigation was conducted on the setting where the demonstration policy is available to the learning algorithm. Under the assumption that both the state space and the action space are discrete, the algorithm solves a minimax optimization to find the reward function. More recent studies tend to focus on the case where only trajectories are available. The algorithms vary widely but they can be roughly categorized into two classes. One architecture is to update the policy function and the reward function coordinately. In each policy update, the policy is optimized for one step to maximize the value of the current reward. In each reward update, the reward function is optimized for one step to ensure that the expert demonstration is optimal under the reward function. The other approaches learn the function approximation of the action-value function, and then test, for each state-action pair in the demonstration, if all the alternative feasible actions lead to values that are no greater than the expert action does (Hester et al. (2018)).

We extend the IRL to the vector-valued case and argue that the extension makes relatively weak assumptions than the existing methods and is hence desired. First, we note that in both categories the algorithms make the assumption on the optimality of the demonstration policy (Ng et al. (2000); Abbeel & Ng (2004)). Such optimality can be furthered formulated as the $\not<$ relationship between

the value of the expert policy and the value of the alternative feasible policy. Hence, when both sides of the value are extended to the vectored case, $\not\geq$ imposes weaker assumption than the $\geq$ of either the scalarized value functions, or the case that $d = 1$. In fact, the element-wise $\geq$ relation of any dimension is sufficient to achieve the $\not\geq$ relation. Intuitively, any non-dominated policies are feasible under the assumption, which allows arbitrary scalarization function as long as they are monotonic. To highlight the significance of the difference in both the assumptions, we note that it is common that existing IRL approaches relax the assumption by assigning a penalty on the violation of the assumption. Such relaxation will drive the actual algorithm from its theory and motivation, while it can be avoided by using vectored value.

## 2.3 VALUE FUNCTION APPROXIMATION

We discuss the approximation of both the state-value function and the action-state value function which is defined as

$$Q^\pi(s,a) = \mathbb{E}_{s'\sim\mathbb{P},s\sim\rho_0(s)}[\sum_{t=0}^{\infty}\gamma^t r_t|s_0 = s, a_0 = a, \pi], \tag{4}$$

and by definition we have $v^\pi(s) = \mathbb{E}_{\pi(a|s)}[Q^\pi(s,a)]$. Without further specification, the discussion applies to $r \in \mathbb{R}^d$ as is defined previously. The approximation is based on the Bellman equation, which describes the recursive relation

$$Q^\pi(s_t,a_t) = \mathbb{E}_{\mathbb{P}(s_{t+1}|s_t,a_t)}[r_t + \gamma\mathbb{E}_{\pi(a'|s_t)}Q^\pi(s_{t+1},a')]. \tag{5}$$

We have then $\pi \in \Pi$ if and only if that whenever $a'$ is feasible under state $s_{t+1}$

$$Q^\pi(s_t,a_t) \not\geq \mathbb{E}_{\mathbb{P}}[r_t + \gamma Q^\pi(s_{t+1},a')]. \tag{6}$$

The inequation is not tractable in general, but we describe the previous studies that addresses the case of $d = 1$ below and present our method in the next section.

The approximate action-state value function can be learned by parametrize the $Q(\cdot)$ function and minimize the discrepancy between both sides of equation equation 6, known as Q-learning (Watkins & Dayan (1992); Sutton et al. (1998); Mnih et al. (2015); Dabney et al. (2017)). Specifically when $d = 1$ and $\not\geq$ degenerates to $\geq$, the discrepancy can be quantified as the Bellman error

$$\frac{1}{2}(Q^\pi(s_t,a_t) - \max_{a'}\mathbb{E}_{\mathbb{P}}[r_t + \gamma Q^\pi(s_{t+1},a')])^2. \tag{7}$$

Combined with the fact that the action-state value function is parametrized (e.g. approximated by a neural network), the Bellman error can be minimized by running optimization program such as SGD. The Q-learning can be conducted off-policy as described.

The optimization in Q-learning is unstable in general and we borrow the techniques and tricks from Mnih et al. (2015) in the implementation. We re-state the techniques for completeness of our presentation. First, the order of expectation and maximization are swapped, resulting in the estimator $\frac{1}{2}(Q^\pi(s,a) - (r_t + \gamma\max_{a'}Q^\pi(s',a')))^2$ which is biased but easier to compute. It then replace the $\max_{a'}Q^\pi(s',a')$ term by $\max_{a'}Q^\pi(s',a'|\theta^-)$, where $\theta^-$ is the parameter of the function approximation of a previous iteration. The tweak aims to reduce the instability due to the correlation of both the approximation terms in equation 7. Third, the reward is clipped and normalized to ensure that the value function is bounded. Note that our algorithm is compatible with improvements over Q-learning (Hasselt (2010); Schaul et al. (2015); Dabney et al. (2017)) and those methods can be employed to increase the quality of the estimation.

## 3 METHODS

### 3.1 ESTIMATING RANGE OF THE VALUE FUNCTION

**Theorem 3.** $\{v^\pi(s)|\pi \in \Pi\} \subseteq conv\{\{v^\pi(s)|\pi \in \Pi_{e_1}\},\ldots,\{v^\pi(s)|\pi \in \Pi_{e_d}\}\}$, *where conv denotes the convex hull and* $e_i$ *is the vector with the i-th element equal to one and zero otherwise. The equality holds when each of the set* $\{v^\pi(s)|\pi \in \Pi_{e_i}\}$ *has an unique element.*

*Proof.* Without loss of generality we assume that the reward $r \geq 0$. If the uniqueness holds, for any element in $\{v^\pi(s)|\pi \in \Pi\}$, there is an explicit $\phi$ which satisfies $\|\phi\|_1 = 1, \phi \geq 0$ such that the element is an optimal value under the reward $\phi^T r$. Hence

$$\text{conv}\{\{v^\pi(s)|\pi \in \Pi_{e_1}\},\ldots,\{v^\pi(s)|\pi \in \Pi_{e_d}\}\} \subseteq \{v^\pi(s)|\pi \in \Pi\}.$$

We proof the reverse by contradiction. Suppose there exists an $\pi \in \Pi$ such that $v^\pi(s) \notin \text{conv}\{\{v^\pi(s)|\pi \in \Pi_{e_1}\},\ldots,\{v^\pi(s)|\pi \in \Pi_{e_d}\}\}$. By separation theorem there exists a $\phi$ such that $\phi^T v^\pi(s) > \phi^T z$ for any $z \in \text{conv}\{\{v^\pi(s)|\pi \in \Pi_{e_1}\},\ldots,\{v^\pi(s)|\pi \in \Pi_{e_d}\}\}$. Let $\phi_{-i}$ be $\phi$ except that the $i$-th element is replaced by zero. If any element of $\phi$ is positive (assume it is the $i$-th without loss of generality), we pick from $\Pi_{e_i}$ the element $z$ with the largest $\phi_{-i}^T z$. The $i$-th element of $v^\pi(s)$ must be greater than that of $z$, which contradicts with the definition of $\Pi_{e_i}$. If otherwise all elements in $\phi$ are negative, $v^\pi(s)$ is dominated by its projection on $\text{conv}\{\{v^\pi(s)|\pi \in \Pi_{e_1}\},\ldots,\{v^\pi(s)|\pi \in \Pi_{e_d}\}\} \subseteq \{v^\pi(s)|\pi \in \Pi\}$, which is the value of a mixed policy. It then contradicts with the optimality of $\pi$. The theorem follows. $\square$

Theorem 3 provides a lower bound of the distance to the range of the value function. The bound is also shown to be tight when the optimal policy under the scalar reward $e_i^T r$ is unique, and the uniqueness commonly happens in practice. Armed with the theorem, estimating the range of the value function amounts to estimating those extreme points. As we have discussed in the Bellman feasibility equation 6, the one-hot vector $e_i$ degenerates the setting to Q-learning with scalarized rewards and the value can be efficiently approximated. We estimate $d$ action-state value functions in our algorithm and denote them as $Q^1(\cdot),\ldots,Q^d(\cdot)$ for further use. The exact function approximator and training process largely depends on the environment and we discuss those technical details in the experiments.

## 3.2 DISTANCE MINIMIZATION

We compute the distance between the value of the alternative policies and the range of the value function, which measures the optimality of the demonstration. We also compute the direction of projection used to measure the distance which describes the combination of the reward vector. Let $Q = (Q^1,\ldots,Q^5)$ be the vectorized action-state value function and let $y$ be the projection of direction. Also denote $\mathcal{T}$ to be the set of state-action pairs $(s,a)$ appeared in the trajectories. We assume that the actions are conducted in a way that the incurred value is not dominated by an alternative feasible action $a'$. Such optimality is learned by maximizing the distance between the set of optimal values $\text{conv}\{\Pi_{e_1},\ldots,\Pi_{e_d}\}$ and the value of the alternative $Q(s,a')$.

The distance is measured by the Euclidean distance in the projected direction, but it could be extended to other measurements in general. It is

$$\sum_{(s,a)} \left[ \max(0, y^T Q(s,a) - \max_{a' \in \mathbb{A}(s)\backslash a} y^T Q(s,a')) \right]. \tag{8}$$

It is worth note that as we do not impose any scalarization on the vectored reward, the model assumption is easier to be satisfied. Consider that the diversity of actions origins from both the diverse reward function and the suboptimality in the actions (such as act randomly), we add a term $c(y^T Q(s,a) - \max_{a' \in \mathbb{A}(s)\backslash a} y^T Q(s,a'))^-$ to make the algorithm more robust. With $c = 0$, it degenerates to the algorithm that only considers the distances. The minimization is reformulated into the following linear program

$$\begin{aligned}
\underset{y,\xi,\eta}{\text{minimize}} \quad & \sum_{(s,a)} c\eta_{s,a}^- - \xi_{s,a}^+ \\
\text{subject to} \quad & \eta_{s,a} \geq y^T(Q(s,a) - Q(s,a')) \geq \xi_{s,a}, \\
& \forall (s,a) \in \mathcal{T}, a' \in \mathbb{A}(s)\backslash a, \\
& y \geq 0, \|y\|_1 \geq 1.
\end{aligned} \tag{9}$$

We describe the complete algorithm in 1, including the range estimation and the distance minimization. In algorithm 1, line #4-12 are the Q-learning technique that computes the vertices of the

---

**Algorithm 1** MMBM

---

1: **Parameters:** learning rate $\alpha$, discount factor $\gamma$, suboptimality factor $c$.
2: **Initialization:** initialize function approximation parameters $w^i$ randomly, $i = 1, \ldots, d$.
3: **Input:** set $\mathcal{T}$ of trajectories;
4: **for** $i = 1$ **to** $d$ **do**
5:     **for** $t$ **to** size of $\mathcal{T}$ **do**
6:        receive the reward $r_t$;
7:     **end for**
8:     **repeat**
9:        Compute the Bellman error $L_1^i$ in equation 7;
10:       Update $w^i$ via gradient-based methods;
11:     **until** convergence of the $i$-th element of $Q(s,a)$;
12: **end for**
13: **for** $t$ **to** size of $\mathcal{T}$ **do**
14:     Compute $Q(s,a) = (Q^1(\cdot), \ldots, Q^d(\cdot))$;
15: **end for**
16: Calculate the constraints $Q(s,a) - Q(s,a')$;
17: Find $y$ by solving the linear program equation 9;
18: **Output:** $Q$ and $y$;

---

superset of the desired value range. Line #13-16 are the estimating of the range of the value function, which is used as the lower bound of the distance. Line #17-18 finally solves the linear program and finds the projection vector $y$ which describes the distance and optimality of the reward space. With our two-step approach, MMBM takes the history of state-action pairs as input, which is usually logged during the gameplay. It solves $y$, which is a quantitative description of agents' motivation and the value structure. Both the part #4-12 and the part #13-18 are generalized and are compatible with other approaches, such as recent advancement on Q-learning or other parametrization and measurement of the distance between $Q(s,a)$ and $Q(s,a')$.

## 4 EXPERIMENTS

We highlight that MMBM possesses several merits of IRL algorithms which enables the following analysis over the behavior and motivation of WoW players. First, the algorithm takes trajectories as input and does not query the policy. It allows us to process recorded data. Also, MMBM does not query the environment dynamic which reduces the computational cost especially when the environment is complex and massive. Third, the algorithm is naturally extended to model the collective behavior of a group when $\mathcal{T}$ is composed of trajectories from multiple players. That helps the model to generate many interesting results as shown below.

### 4.1 IMPLEMENTATION DETAILS

We test MMBM on the WoWAH dataset (Lee et al. (2011); Bell et al. (2013); Shen et al. (2014)), which is an interesting dataset recording a significant amount of gameplay data with over 70,000 players' movements (regarded as actions) from realm *TW-Light's Hope* spanning for the 3-year period. We treat each player as a human agent who conducts an action at each time interval. All available data such as current level or joining the *guild* are treated as observations. The players' trajectories are composed of a sequence of movements (actions) and observations (states), which partially reflect their playing strategies. The oracle $r$ is constructed based on the Yee's research and other WoW case studies (Ducheneaut et al. (2006); Nardi & Harris (2006)). It is illustrated in Tbl. 3 in the appendix.

As discussed before, the function approximator which parametrizes the action-value functions is largely depending on the environment. Taking our experiments on the WoWAH dataset as an example, the Q-network architecture is designed according to the available observations and is applied to all reward signals $i = 1, \ldots, d$. In the network, the categorical elements of the input (e.g. *race*, *class*, etc.) are first processed by an embedding layer, while the numeral elements (e.g. session length, current level etc.) are first fed into a fully connected (FC) layer with rectifier non-linearity. The

output of the embedding layer and FC layer are then concatenated and fed into another FC layer with rectifier non-linearity. A final FC layer is applied to compute the $Q(s, a)$ value for each action $a \in \cup_s \mathbb{A}(s)$.

Our implementation can be viewed as a two-step workflow: The first step, known as Q-learning, estimates the value functions of the players at different states of the gameplay environment; The second step, an extension of the IRL algorithm, that estimates the optimality of the value function learned at the first step. In essence, the two-step methodology decodes the complex interactions between the players and the game environment in the form of the players' reward function. An intuitive illustration of the framework is shown in Fig. 3 in the appendix.

Our code will be publicly released along with this paper.

## 4.2 SOLUTION ON DIFFERENT TRAJECTORY SETS

We present our experimental results of calculating the direction $y$ that most significantly separates the demonstrated behavior and the alternative actions. Recall that $y$ is solved from the linear program equation 9 and note that larger element in $y$ infers relative larger importance of the corresponding element (but it is not linear). We use trajectories that are randomly drawn from specific subsets to compose the constraints of equation 9. We compare the results for different player groups, as shown in 2. Significantly value structures difference is observed between the players at a higher level ($\geq 50$) versus the players at a lower level ($\leq 49$), where the players at the lower level are much more motivated to advance as indicated by the larger component on the Advancement motivation. It also shows interesting difference among players in different classes, *Warrior*, *Hunter*, and *Priest*, where the *Warrior* players value more on Advancement and the *Priest* players value more about relationship. It agrees with the common knowledge in WoW that the spells of *Priest* focus on benefiting (healing, buffing, etc.) the team rather than those of *Hunder* and *Warrior* whose spells are more related about damage, and damage/tank, respectively. Lastly, the results also show that players in the *guild* value more about Teamwork and Relationship motivations as compared to the players that are not in a *guild*. The difference of $y$ are distributed into *advancement* and *escapism* instead. Interestingly, those quantitative results agree with previous knowledge-based studies Ducheneaut et al. (2006); Nardi & Harris (2006) on WoW. Note that the $y$ vector only demonstrates the direction of the projection hence we normalize the vector to $\|y\|_1 = 1$ in the figure

## 4.3 PREDICTING ACTIONS AND VALUES

We evaluate the performance of MMBM in terms of the accuracy of action prediction and the relative accuracy of the estimated value compared with the ground truth. For prediction, the trained model is required to output the next action of the player, given the current state. To mitigate the noise and the short interval in the dataset, the prediction is considered correct as long as the predicted action is taken within five timesteps. The inverse learning error measures the relative error occurred in the value function estimation, defined as $y^T |\hat{v} - v| / |y^T v|$ where $\hat{v}$ denotes the estimation and $v$ the ground truth. As the observed reward is always vectorized, we treat each model's scalarization or projection as if it is correct and count only the error due to value estimation. We compare MMBM with those methods that can be implemented on WoWAH, which require them to work without querying the environment and the policy.

A close examination of the errors that are made during the prediction yields some interesting understandings. As MMBM assumes that every player tries to maximize their value in the IRL algorithm, i.e., everyone is regarded as a rational and optimal player. The assumption is relatively weak enough to include a wide range of diverse behavior, though, MMBM will not be able to distinguish whether a particular action that deviates from an average one is caused by the player's actual intention or the player's sub-optimality during the gameplay. For instance, some of the players could spend hundreds of hours on solo *quest* but fail to level up quickly. This could be due to their intention of enjoying doing the quest repeatedly or the players not knowing the optimal strategy to level up. Considering humans' different ability or skill levels may help address this limitation of our method and we leave it for future works.

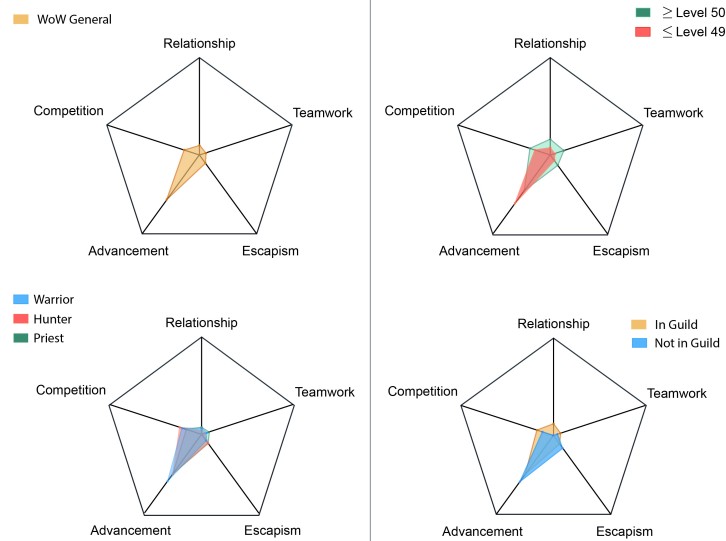

Figure 2: Spider maps to represent player reward mechanism or value structure. **Top-left:** the direction *y* of different motivations for the entire WoW player community; **top-right:** different value structures between the players at higher level ($\geq 50$) and the players at lower level ($\leq 49$); **bottom-left:** different value structures between the players in different classes *Warrior*, *Hunter*, and *Priest*; **bottom-right:** comparison of different value structure of the players who are in *guild* and those who are not in a *guild*

Table 2: Accuracy and error of different approaches

| Approach | Accuracy (%) | ILE (%) |
|---|---|---|
| Proposed approach ($c = 0.5$) | 85.5 | 13.2 |
| Proposed approach ($c = 0$) | 84.5 | 11.0 |
| Linear scalarization Ng et al. (2000) | 48.6 | 21.0 |
| Max entropy scalarizationZiebart et al. (2008) | 54.9 | 22.1 |
| Large margin scalarizationParameswaran & Weinberger (2010) | 59.2 | 19.6 |
| Policy imitation | 36.0 | N/A |
| Proposed approach ($Q()$ learned from Monte-Carlo sampling) | 75.1 | N/A |

## 5 CONCLUSIONS AND OPEN PROBLEMS

We present MMBM, a general IRL model that takes multifaceted human motivations into consideration. The algorithm relies on relatively weak assumptions and does not require any explicit scalarization of the vector-valued reward function. Instead, it leverages the Pareto frontier of the value to characterize the set of optimal policies and to measure the optimality of the recorded behavior. While the algorithm is not relying on the access of policy function nor the dynamics of the environment it can be applied to study complex, interactive environments with its historical dataset. Our experiments on the WoWAH dataset shows the reasonable analysis on the reward mechanism of the players and also the improved prediction accuracy and inverse learning error. We view our work as one of the first that can combine the richness of psychological and game research theories with the rigorousness of RL models. Our goal is beyond winning and losing: Not to simply create software agents that beat human in various games or competitions, but to propose methods that can help to understand the intricacy and complexity of human motivations and their behaviors. We hope to inspire more researchers to investigate this topic further.

Among the challenges in IRL, the vectored reward is the one that has been long open. MMBM addresses this challenge by incorporating the estimated Pareto frontier of the range of the value function, but the algorithm heavily relies on the exactness of the reward function. When any element of the reward function is inaccurate, our algorithm is plausible to incur larger error than those who scalarize the vectored reward, as it is not possible in MMBM to adjust the weight of the reward to mitigate the effect of the inaccuracy. The drawback can be effectively solved if the algorithm is robust with to the reward vector instead of treating it as an oracle. We leave the problem open for further research.

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

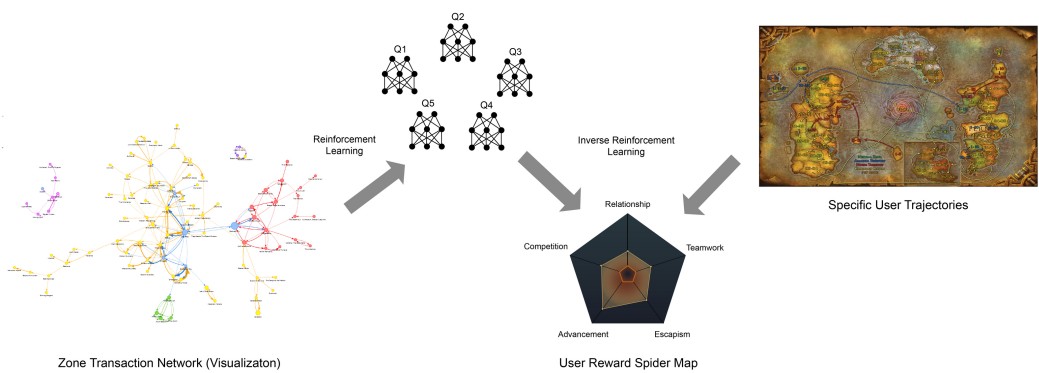

Figure 3: Illustrative execution of Alg. 1 on WoWAH

Table 3: Different types of motivations in WoWAH and corresponding definitions

| Motv. | Category & Definition |
|---|---|
| $r^1$ | **Advancement** describes how fast the player levels up in the game. It's the speed the user levels up, divided by the averaged speed at the entire WoWAH. |
| $r^2$ | **Competition** describes if the player joins *Battleground* or *Arena* and competing with human opponents. It equals the number of visits. |
| $r^3$ | **Relationship** is linear to the duration that the player has been in the current *guild*. |
| $r^4$ | **Teamwork** describes the intention of conducting teamwork, which is the number of recent zones with teamwork features visited. Zones with teamwork features include *Battleground*, *Arena*, *Dungeon*, *Raid*, or a zone controlled by *The Alliance*. |
| $r^5$ | **Escapism** is the linear combination of the duration of the recent game session and the number of days the player continuously login to the game recently. |

## A  THE ORACLE $r$ IN THE EXPERIMENTS

We attach the vector-valued reward, as an oracle, used in our experiments in Tbl. 3. We note that the vectors should be treated as known and bounded and shall be generated by either collecting observations or applying domain knowledge.

## B  DYNAMICS OF THE HUMAN MOTIVATION

The motivation of gameplay may evolve. It can also be impacted by the new design or new versions of the game environment. We investigate how would a design update affect the players' motivations and behavior and how we can quantify this impact? To achieve this, we retrain the linear program 9 with linear constraints random drawing from specific time ranges. With the time range moving chronologically, we show the evolution of game motivation characterized by the elements in *y*. Fig. 4 illustrates the trend of *Advancement*, *Competition*, *Relationship*, *Teamwork*, and *Escapism*. We observed the dramatic increase in Advancement and Competition during the mid-to-late period on the graph. It happens at around the 150000th time interval, which coincides with the release of the patch *Wraith of the Lich King* on November 2008. Analyzing the game update patch, two primary reasons can explain the increased level of motivation on Advancement. First, the patch increased the maximum player level from 70 to 80. As a result, the players with level 70, the previous max level, were rushing to complete the remaining ten leveling ups to reach the new max level. Second, the patch introduced two new classes in the game, namely *Death Knight* and *Shaman*, and this gave incentives to many players to open the secondary accounts and to level up them is the first thing to do afterward. Meanwhile, the reason for more Competition is that many players tend to join player-

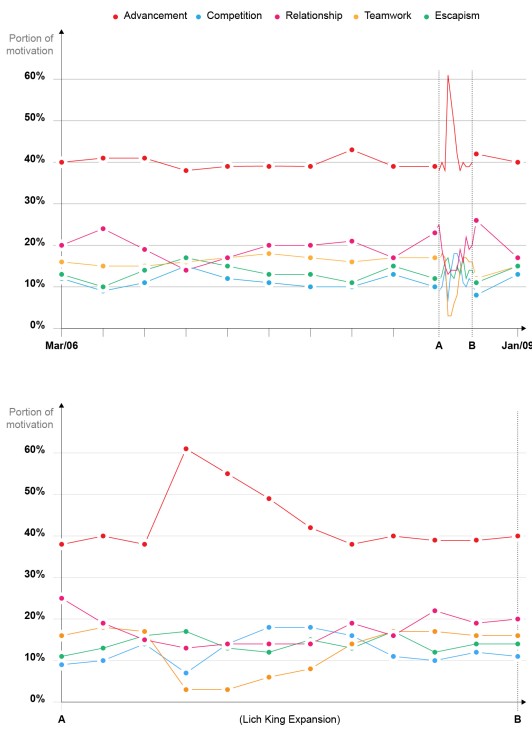

Figure 4: **Top**: trends of different kinds of motivations during from Mar 2006 to Jan 2009; **Bottom**: the enlargement of the top figure during around the release of patch *Wraith of the Lich King*

versus-player (PvP) to compete with other human players to get more familiar with the mechanism of their new avatar. It is also noticed that the satisfactions are not independent of each other: players spend more time on advancement usually have insufficient time to complete tasks which require teamwork but provides no experience for leveling up. As shown in Fig. 4, that the component for teamwork decreases each time the component for advancement increases, and vice versa.

We also analyze the overall trend of the game during the three years when WoWAH was collected. It turns out that the game emphasis more on teamwork and relationship during the period, partially because the dataset was collected only two years after the game release, and the players are getting more and more involved in the game during that time. Apart from that, the components of different kinds of motivations are under influences from both game patches and updates, and game user community. Overall, our algorithm provides insights in Fig. 4 and the analysis that may be inspiring for researchers and game researchers.

