# OpenReview forum: "Beyond Winning and Losing: Modeling Human Motivations and Behaviors with Vector-valued Inverse Reinforcement Learning"
_ICLR.cc/2019/Conference_

### Official Review · AnonReviewer1 · 2018-11-02
**Review of "Beyond Winning and Losing..."**

**Rating:** 4
**Confidence:** 4

**Review:**

======== Summary ============

The authors consider a setup where there is a set of trajectories (s_t, a_t, r_t) where r_t is a *vector* of rewards. They assume that each agent is trying to maximize \sum_t \gamma^t (\phi . r_t) where \phi is a preference vector that lives on the simplex. Their goal is to calculate \phi (and maybe also an optimal policy under \phi?). The

The authors first prove that this problem can be decomposed into finding Q functions for optimal policies for each component of r_t individually, and then solving for \phi that rationalizes the trajectory of actions in terms of these Q functions. Given the entire collection of trajectories, they perform off-policy Q-learning on each component of r_t in order to learn the Q function for that component, and then use linear programming to solve for \phi based on these Q function.

========== Comments =============

I think it's a worthwhile direction to combine IRL with modeling a diversity of preferences among agents. I can imagine several reasons you might want to do this, but the authors are not clear what their goal is besides "to propose methods that can help to understand the intricacy and complexity of human motivations and their behaviors". Is the goal to do better policy prediction? To do better policy prediction conditional on \phi? To infer \phi to understand people's preferences from a social science perspective? These all seems reasonable but not sufficiently teased out in the work. (For comparison, IRL is typically - although not always - interested in learning the reward function in order to construct robust policies that maximize it). The authors also don't seem to solve a particular task of importance on the WoW dataset.

The theoretical approach seems sound, and I liked the way their algorithm was motivated and the way the problem was decomposed into off-policy Q-learning and then solving for \phi.

However, I found myself quite confused in the experimental section (4.3). The authors evaluate their approach by action prediction. Given the trajectories, is \phi computed for each player and then compute actions based on that value of \phi? Is \phi computed on the same trajectory data used for evaluation or a different subset? Or is action prediction performed in aggregate across the entire population? The experimental setup was never clarified for this (main) experiment.

I was also confused about the motivation for Figure 2 and Appendix D. The authors are showing that their predictions about which reward is motivating the players is consistent with external factors. But wouldn't you see the same thing if you just plotted the observed *rewards* themselves? E.g. players in a guild will achieve more Relationship reward.
The proposed approach takes the vector of reward, learns which actions are consistent with achieving each reward, then infers from the actions which reward is trying to be achieved. What advantages does this have vs. just looking at the empirical trajectory of rewards for each player/group?
I can certainly imagine that the IRL approach has certain advantages over looking at the empirical reward stream, but the authors have not talked about this nor compared against it experimentally.

The writing could also use some improvement for a future iteration, I've listed a few points below:

pg.1, Neither Brown & Sandholm nor Moravcik et al use "RL algorithms"
pg.1, Finn et al unmatched )
pg.1, "a scalar reward despite observed or not" -> "a scalar reward whether observed or not"
pg.2, "Either the range of" -> "Both the range of" (and this sentence needs further cleanup)
pg.2, "which records the pathing of players" ??
Theorem 3: "each of the set e_i has an unique element..." This isn't clear. I think you mean "For each e_i there is a unique vector v^\pi(s) for all \pi \in \Pi_{e_i} . The equality holds if these vectors are distinct for each e_i".
pg. 5 "If otherwise all elements in \phi are generative" how can they be negative if they are on the simplex?
pg.5 "we do not perform any scalarization on the reward...the model assumption is easier to be satisfied" I think this is a strange comparison to IRL because in IRL you're trying to find a (possibly parametric) function (s,a) -> R, whereas here you're *given* the vector R and are trying to find \phi. So while you have more degrees of freedom by adding \phi, you lose the original degrees of freedom in the reward function.

---

> ### Author Response · Authors · 2018-11-12
> **Response to the review**
>
> Thanks a lot for your detailed review! A quick note is that pg. 5 "If otherwise all elements in \phi are generative" the \phi is the existence specification by the separation theorem. It is not what mentioned in Section 3.2 and is not restricted to the simplex. We are updating Section 3.2 to avoid the possible ambiguity. We also note that the model is trained and tested on disjoint subsets of WoWAH. The spider map was calculated by the entire population though.
>
> For the contribution of the paper, we believe it is indeed worth investigating combining IRL with the diversity of preferences among agents. In fact, the problem of IRL with scalar-valued reward has been long open. The assumption (that is used by almost every IRL algorithm) is too strong to complex agents (such as humans). We developed Theorem 3.1 which significantly weaker the assumption. We agree that armed with the theorem, Section 3.2 and Section 4 was not aiming at a clear objective as you may expect. To make the objective more clear, it is more reasonable to reproduce the policy (or the set of policy) that was used to generate the trajectory dataset. That is because we already have the reward vector and also the Pareto optimal relationship assumption, and the policy is the only unknown element. Some updates on the algorithm will be necessary then, which is currently undergoing.
>
> For experiments, we agree that it can be confusing to demonstrate the real-world problem. There are several constraints to run the algorithm on the real-world dataset, such as the query of the state transition function. That makes the experiments itself more dependent on its context. As a solution, we find it better to add some well-known experiments such as openAI gym or gridworld with vector-reward, which provide a more intuitive understanding of the empirical performance of the algorithm. We are currently working on that.
>
> We have updated the writing in the revised version of the paper. Thanks a lot for pointing them out!

---

### Official Review · AnonReviewer3 · 2018-11-03
**Interesting work, but need further improvement**

**Rating:** 4
**Confidence:** 4

**Review:**

This paper presents NMBM, a general inverse reinforcement learning (IRL) model that considers multifaceted human motivations. The authors have motivated and proposed the algorithm (Section 2 and 3), and demonstrated some experiment results based on a real-world dataset (WoWAH, Section 4).

-- Originality and Quality --

To the best of my knowledge, the proposed NMBM algorithm is new. However, I feel that the derivation of this algorithm is relatively straightforward based on existing literature. Specifically, this algorithm is based on (1) Theorem 3 and (2) the linear program defined in equation 9. My understanding is that both Theorem 3 and the derivation of the linear program in equation 9 are relatively straightforward based on existing literature.

On the other hand, the experiment results in Section 4 are very strong and interesting. It is the main strength of this paper.

-- Clarity --

My understanding is that the writing of Section 3 and 4 can be (and should be) further polished.

Some key notations in the paper seem to be wrong:

(1) In Theorem 3, how can the value function v^\pi(s) be in the convex hull of policies? Also, e_i is not a set.

(2) In equation 9, the linear program, \eta should be another decision variable.

-- Pros and Cons --

Pros:

1) Strong experiments.

Cons:

1) Insufficient novelty for algorithm design.

2) No performance analysis for the proposed algorithm.

3) Clarity needs to be further improved.

---

> ### Author Response · Authors · 2018-11-11
> **Response to the review**
>
> Thanks very much for the review! For the algorithm, we agree that it is indeed straightforward. We would like to note, though, that dealing with the scalarized reward function has long been an open problem in inverse reinforcement learning. We have tried other (more complex) approaches but finally found out that the lower bound introduced in Theorem 3.1 is the most appropriate one. We believe that the estimation in Theorem 3.1 is a reasonable solution to the problem. On the other hand, there are rooms for other subtle methods related to distance measurement, in Section 3.2. We are working on employing that into the algorithm for better performance.
>
> We thank the reviewer for the comments on the strong experiments. In fact, involving some real-world problem gives the implication beyond the typical simulator-based environments. It is important that the reviewer mentions "No performance analysis for the proposed algorithm". Keeping that in mind, we are working on adding some results on openAI gym, which includes benchmarked tasks and quantitative evaluations.
>
> We have corrected the notation typos in the updated draft and updated some other writings for its clarity.

---

### Official Review · AnonReviewer2 · 2018-11-06
**interesting paper with some issues**

**Rating:** 5
**Confidence:** 3

**Review:**

This paper studies inverse reinforcement learning with a vector-valued setting. A key motivation of the paper, as suggested by its title, is to incorporate and analyze the complex human motivations.

The proposed setting seems new to me, although vectored-valued rewards and Pareto optimality have been studied in the context of RL. The biggest issue of this paper, in my opinion, is it doesn't properly support its claim that it improves the understanding of the agents' motivations and the reward functions. Details comments / questions are listed below.

- Pareto dominance is a rather weak relation. When the number of criteria increases, it is less likely one alternative dominates another. In this case, the binary comparisons defined in Sec. 2.1 becomes less discriminative. Is this a problem to the proposed method?

- Pareto dominance and vector-valued rewards have been studied in preference-based reinforcement learning, such as Fürnkranz et al. 2012 @ MLJ and Cheng et al. 2011 @ ECML.

- Please fix the citation style in the paper and use \citep and \citet properly.

- The empirical study in this paper doesn't properly support the authors' claim. (1) It's questionable to assume the actions of a player in an online game are optimal or even rational. (2) The results presented in Figure 2 is hard to read and the differences look minor. (3) Maybe I miss it, but has Table 2 been referenced and explained in the paper?

---

> ### Author Response · Authors · 2018-11-12
> **Response to the review**
>
> Thank you for the review! The most important clarification that we would like to make, is that "Pareto dominance is a rather weak relation" makes the model rather strong. That is because the dominance relation is the assumption of the IRL models, and weak assumptions are desired. We believe that justifies our motivation of studying the Pareto dominance in the IRL regime.
>
> On the empirical study, we agree that presenting the algorithm on only the real-world environments may depend on the rational assumption. In fact, we are aware of that the diversity in action originates from both the diversity of the agents' objective and their optimality (or even rationality). There is not too much one can resolve in the real-world dataset, but we can test the algorithm on the well-known RL environment and show the performance. We are also working on improving the writing quality and thanks for pointing those out.

---

### Meta-Review · Area_Chair1 · 2018-12-14

**Confidence:** 3
**Recommendation:** Reject

**Metareview:**

Pros:
- new multi-objective approach to IRL
- new algorithm
- strong results
- real-world dataset

Cons:
- straightforward theoretical extensions
- unclear motivation
- inappropriate empirical assessment metrics
- weak rebuttal

All the reviewers feel that the paper needs further improvements, and while the authors comment on some of these concerns, their rebuttal and revised paper does not address them sufficiently. So at this stage it is a (borderline) reject.